# Evaluating the effect of Bolsa Familia, Brazil's conditional cash transfer programme, on maternal and child health: A study protocol

Ila Rocha Falcão[1,2]*, Rita de Cássia Ribeiro-Silva[1,2], Flávia Jôse Oliveira Alves[2,3], Naiá Ortelan[2], Natanael J. Silva[2], Rosemeire L. Fiaccone[2,4], Marcia Furquim de Almeida[5], Júlia M. Pescarini[2,6], Cinthia Soares Lisboa[2,7], Elzo Pereira Pinto Júnior[2], Enny S. Paixao[2,6], Andrea J. F. Ferreira[2,3], Camila Silveira Silva Teixeira[2,3], Aline dos Santos Rocha[1,2], Srinivasa Vittal Katikireddi[8], M. Sanni Ali[6], Ruth Dundas[8], Alastair Leyland[8], Laura C. Rodrigues[2,6], Maria Yury Ichihara[2,3], Mauricio L. Barreto[2,3]

1 School of Nutrition, Federal University of Bahia, Salvador, Brazil, 2 Centre for Data and Knowledge Integration for Health (CIDACS), Oswaldo Cruz Foundation, Salvador, Brazil, 3 Institute of Collective Health, Federal University of Bahia, Salvador, Brazil, 4 Department of Statistics, Federal University of Bahia, Salvador, Brazil, 5 School of Public Health, University of São Paulo, São Paulo, Brazil, 6 Epidemiology and Population Health, London School of Hygiene and Tropical Medicine, London, United Kingdom, 7 Feira de Santana State University, Feira de Santana, Brazil, 8 MRC/CSO Social and Public Health Sciences Unit, University of Glasgow, Glasgow, Scotland

* falcao.ila@gmail.com

**Data Availability Statement:** All data will be obtained from Centro de Integração de Dados e Conhecimentos para Saúde (CIDACS). Importantly,

## Abstract

### Background

Conditional Cash Transfer Programs have been developed in Latin America in response to poverty and marked social inequalities on the continent. In Brazil, the *Bolsa Familia* Program (BFP) was implemented to alleviate poverty and improve living conditions, health, and education for socioeconomically vulnerable populations. However, the effect of this intervention on maternal and child health is not well understood.

### Methods

We will evaluate the effect of BFP on maternal and child outcomes: 1. Birth weight; 2. Preterm birth; 3. Maternal mortality; and 4. Child growth. Dynamic retrospective cohort data from the 100 Million Brazilian Cohort (2001 to 2015) will be linked to three different databases: Live Birth Information System (2004 to 2015); Mortality Information System (2011 to 2015); and Food and Nutritional Surveillance System (2008 to 2017). The definition of exposure to the BFP varies according to the outcome studied. Those who never received the benefit until the outcome or until the end of the follow-up will be defined as not exposed. The effects of BFP on maternal and child outcomes will be estimated by a combination of propensity score-based methods and weighted logistic regressions. The analyses will be further stratified to reflect changes in the benefit entitlement before and after 2012.

restrictions apply to the availability of these data, which will be licensed for exclusive use in the studies, and are thus not publicly available. Upon reasonable request and with the express permission of CIDACS, the authors are willing to make every effort to grant data availability. No data was used or analyzed for this protocol. Data will be included in completed study.

**Funding:** CIDACS received financial support by MCTI / CNPq / MS / SCTIE / Decit / Bill & Melinda Gates Foundation's Grandes Desafios Brasil – Desenvolvimento Saudável para Todas as Crianças (call number 47/2014) (grant number OPP1142172). CIDACS and the 100 Million cohort received financial support from the Wellcome Trust (grant number 202912/Z/16/Z), the Health Surveillance Secretariat, Ministry of Health, Brazil, Bahia State (Decentralized Execution Term – TED number 159/2019), Research Support Foundation of the State of Bahia (FAPESB) (grant number INT0001/2015), the Research and Project Funding Agency (FINEP) (Notice CT-INFRA - FIOESTAT - Agreement number 04.10.0635.00, reference number 811/10). CIDACS received material support (referring to rooms in Bahia Technology Park in Salvador, state of Bahia) from Secretariat of Science and Technology of the State of Bahia (SECTI) (term of assignment of movable property 048/2018, process number 1430150022698). Individual financial support: IRF received a doctoral scholarship from the Research Support Foundation of the State of Bahia (FAPESB) (grant number BOL2330/2016). ESP is a fellow supported by the Wellcome Trust (grant number 13589/Z/18/Z). SVK acknowledges funding from a NRS Senior Clinical Fellowship (grant number SCAF/15/02). SVK and AHL also receive funding from the Medical Research Council (grant number MC_UU_12017/13) and Scottish Government Chief Scientist Office (grant number SPHSU13). The funders had no role in study design, data collection and analysis, decision to publish, or preparation of the manuscript. The authors received no specific funding for this work.

**Competing interests:** The authors have declared that no competing interests exist.

**Abbreviations:** ATT, Average Treatment Effect on Treated; BFP, Bolsa Familia (Family Grant) Program; CadÚnico, Single Registry for Federal Government's Social Programs; CCT, Conditional Cash Transfer Program; CIDACS, Centre for Data and Knowledge Integration for Health; CIDACS-RL, CIDACS Record Linkage; HAZ, Height-for-age z-score; IGD, Decentralized (municipal) management index of the BFP; IPTW, Inverse Probability of Treatment Weighting; LBW, Low birth weight; LGA,

## Discussion

Harnessing a large linked administrative cohort allows us to assess the effect of the BFP on maternal and child health, while considering a wide range of explanatory and confounding variables.

## Background

Poverty and social inequality have been identified as major social causes of poor health, requiring public policies and strategies to eradicate poverty and improve the most vulnerable populations' living and health conditions [1–3]. Despite the advances observed on maternal and child health in the last decades, the slow decline in maternal mortality and the persistence of adverse outcomes, such as preterm birth (PTB), low birth weight (LBW), and child malnutrition, especially among the low and -middle-income countries (LMIC), hinder the achievement of the Sustainable Development Goals (SDGs) [4–8]. In Brazil, the maternal mortality ratio was 59.7 per 1,000 live births in 2015 (a 57% decline in 25 years), and the national prevalence of PTB and LBW were respectively 11.1%, and 8.4% [9]. Malnutrition estimates in children under five years old enrolled in the Bolsa Familia Program (BFP), a Brazilian Conditional Cash Transfer (CCT), showed a high prevalence of stunting (12.7%) and overweight/obesity (18.4%) in 2014 [10].

CCTs have been adopted as a strategy to promote maternal and child health [11–13]. Programs focused on combating immediate and future poverty may improve access to health, education, social assistance, employment, and income [14]. The BFP is one of the oldest and largest CCTs in the world, with over 13.2 million beneficiary families, corresponding to 96% coverage of the country's poor households (estimates for February 2020) [15]. While Brazil was one of the pioneers in implementing the CCT in Latin America, there is still little evidence on the effect of BFP on maternal and child outcomes, especially from studies using robust methods and large-scale individual-level data with an extended period of follow-up [11, 12, 14]. Understanding the health and health equity impacts of social policies is important to inform policymaking, including decisions about ongoing investment in these schemes [12, 14, 16, 17].

The most important contribution of the proposed research will be developing robust evidence of the effect of the BFP on maternal and child outcomes, using a cohort which allow us assessing more robust statistical analyzes in the general population and separately for specific subpopulations.

## Methods

### Primary objective, study design, and overall population

We aim to evaluate the effect of BFP on maternal and child outcomes in the 100 Million Brazilian Cohort [18]. The main objective of the Cohort is to enable the study of the social determinants and the effects of social policies and programs on the different aspects of health in Brazil [18]. It is a dynamic retrospective cohort, the population of which is derived from more than 114 million individual records from the Single Registry for Federal Government's Social Programs (CadÚnico). The cohort contains administrative records from CadÚnico and the BFP Payroll. CadÚnico identifies and characterizes low-income households and registration is required in order to receive any Federal Government's social programs, such as the BFP [19]. The Cohort allows us to extract socioeconomic information from the individual, the

Large for Gestational Age; LMIC, Low and -middle-income countries; PTB, Preterm birth; PS, Propensity Score; SDG, Sustainable Development Goals; SGA, Small for Gestational Age; SIM, Mortality Information System; SINASC, Live Birth Information System; SISVAN, Food and Nutrition Surveillance System; WAZ, Weight-for-age z-score; WHZ, Weight-for-height z-score.

household, and data related to receiving the benefit. The detailed variables and databases to be used are shown in Table 1.

The primary objective will be achieved by linking the Cohort (2001 to 2015) and data from (i) the Live Birth Information System (SINASC) (2004 to 2015); (ii) the Mortality Information System (SIM) (2011 to 2015); and (iii) the Food and Nutrition Surveillance System (SISVAN) (2008 to 2017). We will use CIDACS Record Linkage (CIDACS-RL) to link the databases [20]. The linkage procedures are common for the 100 Million Cohort studies and consist of two stages. The first will be a deterministic linkage, and the second will be based on the similarity index. More detailed information can be consulted in previous publications [21, 22]. The CIDACS-RL is a tool for linking individual records based on identifiers: name, gender, age or date of birth, mother's name, and the municipality of residence [22]. All linking procedures will be performed at CIDACS (Center for Data Integration and Knowledge for Health, Fiocruz) [23] in a strict data protection environment and complying with ethical and legal standards [24].

## The Bolsa Familia Program (BFP)

We describe the policy in accordance with the TIDieR-PHP reporting guideline [25]. The checklist consists of nine items and helps researchers to describe the characteristics of population health and policy interventions. The BFP was implemented from a national decree in 2004, with eligibility criteria (poverty and extreme poverty cutoff points) and incorporation of benefits that varied over time [26–31]. The cut-off points and the eligibility criteria are shown in Table 2. The selection of households eligible for the BFP occurs through enrollment in the CadÚnico [26, 31]. Households served by the BFP receive a monthly cash benefit through a withdrawal card issued by the Caixa Econômica Federal [32].

The BFP is equipped with fraud prevention control mechanisms, with public access to beneficiaries' individual data over the internet and semiannual comparison of CadÚnico's enrolled data with other databases [21]. The suspension of households from BFP can occur due to failure to update the registration information, no longer fitting the profile (eligibility

**Table 1. Structure and main components of the 100 Million Brazilian Cohort, sources of data, and relevant variables.**

| Components | Data source | Period | Number of Records | Relevant variables |
|---|---|---|---|---|
| Cohort Baseline | Single Registry (CadÚnico) | 2001 to 2015 | 114,008,317 | Socioeconomic and demographic conditions (information on family dynamics, childcare arrangements, parental employment, income, housing family formation, dissolution, social programs information, household characteristics). |
| Intervention (Exposure) | Bolsa Familia Program (BFP) | 2004 to 2015 | 27,376,582 | Start and end of data receipt of benefit, total value by family, and number of months received. |
| Outcomes | Live Birth Information System (SINASC) | 2001 to 2015 | 44,485,274 | Characteristics of the newborn (sex, Apgar score in the 1 and 5 minutes, birth weight, presence of an abnormality, congenital anomalies identified at birth), characteristics of the mother (age, marital status, education, race, place of residence), characteristics of pregnancy and delivery (number of previous pregnancies of live births, stillbirth or abortion, gestational age, place of birth, type of delivery, number of fetuses, number of prenatal visits, month that started prenatal). Some variables such as the month in which the woman started prenatal care and gestational age (continuous) are only available for the period from 2011 to 2015. |
| Outcomes | Mortality Information System (SIM) | 2000 to 2015 | 17,829,111 | Type of death, date of death, date of birth, sex, race, education, duration of the pregnancy, single or multiple pregnancies, type of delivery, age of mother, gestational age, birth weight, and death cause. |
| Outcomes | Food and Nutrition Surveillance System (SISVAN) | 2008 to 2017 | 307,245,508 | Date of birth, age, sex, race/ethnicity, traditional communities, anthropometric data (weight and height), measurement date, presence of chronic diseases (diabetes and cardiovascular diseases), and deficiencies and complications (diarrhea and anemia). |

**Table 2. Changes in the eligibility criteria and inclusion of new groups of beneficiaries.**

| Year | Extreme poverty* (R$) | Poverty* (R$) | Inclusion of new groups (varying benefits) |
|------|------------------------|----------------|---------------------------------------------|
| 2004 | 50.00 | 100.00 | No change |
| 2006 | 60.00 | 120.00 | No change |
| 2009 | 70.00 | 140.00 | Concession of benefits to households with adolescents aged 16–17 years enrolled in education institutions |
| 2012 | No change | No change | Concession of benefits to households with children aged zero to six. Concession of varying benefits to pregnant women and nursing mothers |
| 2014 | 77.00 | 154.00 | No change |
| 2016 | 85.00 | 170.00 | No change |
| 2018 | 89.00 | 178.00 | No change |

* Household units with a per capita household income less than or equal to the mentioned value.

criteria), and noncompliance with conditionalities [32]. The program's conditionalities are geared to participation in education, health, and social assistance. In the field of health, conditionalities include actions, such as immunization, prenatal care, and child growth monitoring [26, 31].

## Logic models

We created a logic model to describe the hypothesized mechanisms through which the BFP might affect maternal and child outcomes (Fig 1). The socioeconomic characteristics can influence both the receipt of the benefit and maternal and child health outcomes [33–39].

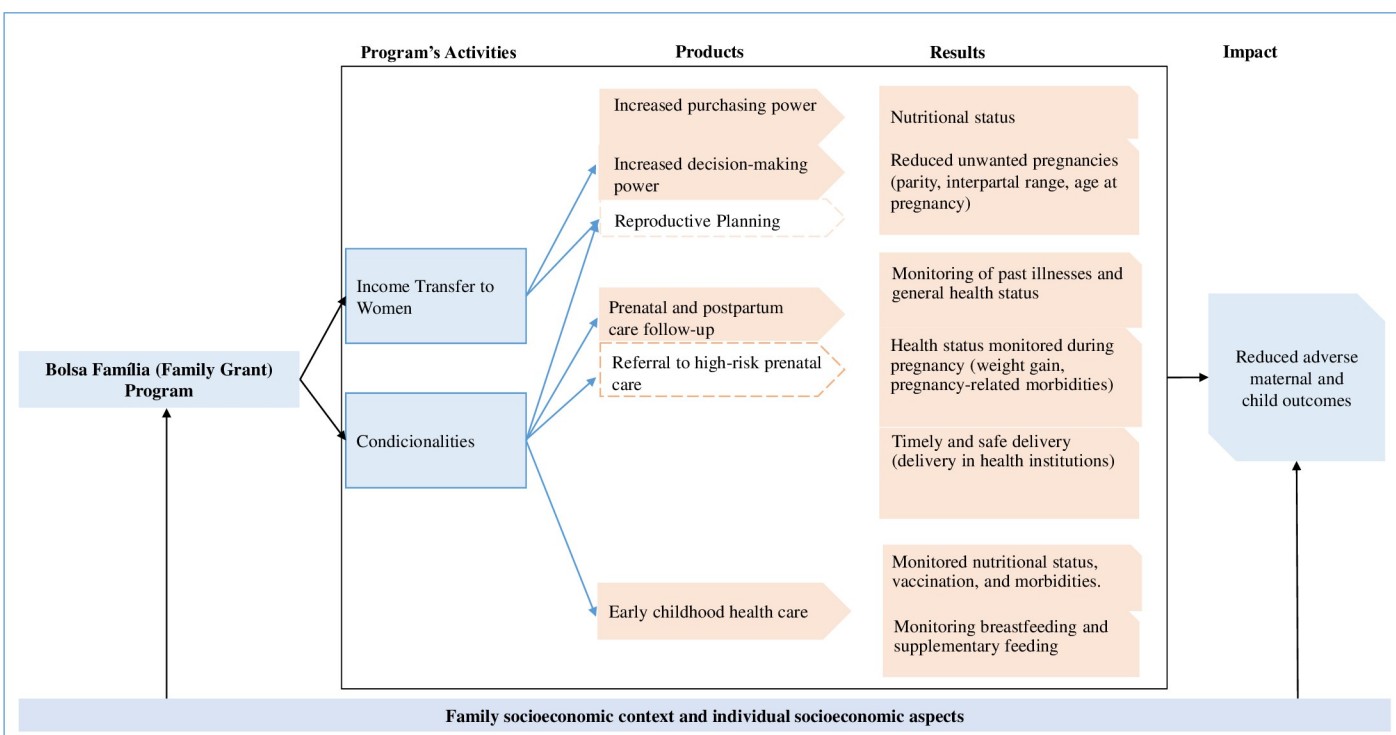

**Fig 1. Logical model of the impact of the Bolsa Familia Program (BFP) in reducing adverse maternal and child outcomes.**

Characteristics of particular relevance include targeting monetary resources preferentially to women and the fulfillment of conditionalities. Despite not being a guarantee, the BFP may increases women's decision-making power [40], has the potential to transform women into heads of households with responsibility for directing the money received. The transfer of income to women can have a more immediate effect on maternal and child health outcomes, with female empowerment, the allocation of money for the purchase of food, and the use of health services [41–46].

On the other hand, BFP also requires the fulfillment of conditionalities, using services during pregnancy, puerperium, and early childhood [26, 31]. Using health services is an important determinant of maternal and child outcomes [4, 47–54] because it can have an immediate effect on these outcomes, with immunization, nutritional counseling and preventive behaviors during prenatal care, monitoring of comorbidities, and connected to the place of birth [47, 55–57]. The reduction of adverse maternal and child outcomes depends on joint efforts that ensure access to quality health services and lower social inequalities [12, 16, 17, 34, 35, 58].

## Secondary objectives, study population, definition of exposure, and outcomes

The definitions of outcomes, study populations, and exposure to BFP will be presented separately (detailed information in Chart 1, as supplementary material), according to the objectives:

- assess the effect of BFP on birth weight, small and large for gestational age (SGA/LGA) and on preterm births

- evaluate the effect of BFP on maternal mortality

- assess the effect of BFP on child malnutrition.

**i) Birth weight, SGA, LGA, and preterm birth.** *Study population*. The study will include baseline data from the "100 Million Brazilian Cohort" linked to SINASC (Table 1). The study population will consist of the first and the second live birth of women registered in the cohort baseline, from 2004 to 2015, with ages ranging from 10 to 49 years. The study population for the SGA, LGA and PTB refers to 2012 to 2015 period, due to the inclusion gestational age as a continuous variable in 2011.

Multiple births and newborns with congenital anomalies will be excluded to avoid bias, as these conditions are known to be strongly associated with low birth weight and PTB [52, 59–61]. Fetal viability criteria can be applied [62–65]. Regarding birth weight, the inclusion of the first live birth is a strategy to capture the effect of receiving the BFP during the first and the second pregnancy. Ordering the live births will allow us to select/extract the population of interest and obtain previous birth information such as inter-birth interval, low birth weight, and preterm birth.

Since nulliparous women are at increased risk of chronic and acute medical and obstetrical complications leading to preterm birth, we will restrict the population related to PTB to singleton live births whose mothers in reproductive age have at least one child before joining the Cohort. Only the firstborn after enrolment will be included.

*Exposure to BFP*. The exposure is defined as having started receiving BFP before the birth of their child in the 2004 to 2015 period (or 2012 to 2015 for SGA, LGA and PTB) and did not stop receiving from pregnancy to delivery. Live births of women who did not receive the benefit at any time until delivery will be considered as not exposed.

*Outcomes.* Birth weight will be considered as (1) birth weight, in grams (continuous variable), and (2) birth weight categorized into very low, low, normal, and high (see Table 3) [66].

**Table 3. Description of the outcomes that will be considered in studies by assessing the impact of the Bolsa Familia Program (BFP).**

| Objective | Original variables used to construct the outcome | Outcome |
|---|---|---|
| To evaluate the effect of BFP on birth weight, small and large for gestational age and preterm birth | Birth weight in grams | Birth weight in grams (continuous variable) |
| | | Adequate birth weight ($\geq$2500g) vs. low birth weight ($<$2500g) |
| | | Adequate weight (2500-3999g) vs. extremely low weight ($<$1000g), very low weight (1000-1499g), low birth weight (1500-2499g) and macrosomia ($\geq$4000g) |
| | Weight in grams and Gestational age in full weeks (available from 2011) | Adequate for gestational age (between 10th and 90th percentiles) vs. Small for gestational age ($<$10th percentile) and Large for gestational age ($>$90th percentile) |
| | | Extreme weights for gestational age: 10th to 90th percentile vs. $<$3rd percentile; 3rd to 9th percentile, 91st to 97th percentile and $>$97th percentile |
| | Gestational age in categories | non-preterm birth ($\geq$37 gestational weeks) vs. preterm birth ($<$37 gestational weeks) |
| | | Non-PTB vs. moderate-to-late PTB (32 to 36 gestational weeks), very PTB (28–31 gestational weeks) e extreme PTB ($<$ 28 gestational weeks) |
| To assess the effect of BFP on maternal mortality | Underlying cause of death | Non-death vs. death of a woman during pregnancy or up to 42 days after the end of pregnancy, due to any cause related to or aggravated by the pregnancy, but not due to accidental or incidental causes. |
| | Intermediate cause of death | |
| To assess the effect of BFP on child malnutrition | Length/height in centimeters, age in months, and sex | Height-for-age z-score (HAZ) |
| | | HAZ $\geq$ –2 (benchmark) vs. HAZ $<$ –2 (stunting) |
| | | HAZ $\geq$ –2 (benchmark) vs. HAZ $<$-3 (severe stunting) and HAZ $\geq$ –3 to HAZ $<$ –2 (moderate stunting) |
| | Weight in grams, age in months, and gender | Weight-for-age z-score (WAZ) |
| | | WAZ $\geq$ –2 to $\leq$ +2 (benchmark) vs. WAZ $<$ –2 (underweight) |
| | | WAZ $\geq$ –2 to $\leq$ +2 (benchmark) vs. WAZ $<$ –3 (severe underweight) and WAZ $\geq$ –3 to $<$ –2 (moderate underweight) |
| | Weight in grams, length/height in centimeters, and gender | Weight-for-height z-score (WHZ) |
| | | WHZ $\geq$ –2 and $\leq$ +2 (benchmark) vs. WHZ $<$ –2 (wasting) |
| | | WHZ $\geq$ –2 and $\leq$ +2 (benchmark) vs. WHZ $<$ –3 (severe wasting) and WHZ $\geq$ –3 and $<$ –2 (moderate wasting) |
| | | WHZ $\geq$ –2 and $\leq$ +2 (benchmark) vs. WHZ $>$ +2 (overweight/obesity) |
| | | WHZ $\geq$ –2 and $\leq$ +2 (benchmark) vs. WHZ $>$ +3 (obesity) and WHZ $\leq$ +3 to $>$ +2 (overweight) |

Small for Gestational Age will be defined as birth weight according to gestational age and gender below the 10th percentile; Adequate for Gestational Age, between the 10th and 90th percentiles; and Large for Gestational Age, above the 90th percentile [66, 67]. Categories will also be explored, including weight extremes for gestational age (Table 3).

Preterm birth will be defined as 1. PTB (22 to <37 gestational weeks) vs. non-PTB (37 to 42 gestational weeks); and 2. stratified (Table 3), according to the degree of severity [66].

**ii) Maternal mortality.** *Study population*. The study will include data from 100 Million Brazilian Cohort linked to SINASC and SIM. The study population will consist of women of reproductive age (10 to 49 years) according the surveillance criteria in Brazil, registered in the Cohort baseline, in their last pregnancy in the 2004 to 2015 period.

*Exposure to BFP*. The exposure is defined as having started receiving the BFP before or during pregnancy and did not stop receiving the benefit before the outcome or until childbirth. Women who have not received the benefit at any time until childbirth or the puerperium will be considered as not exposed.

*Outcome*. Maternal death will be defined as the death of women during pregnancy or up to 42 days after the end of pregnancy, due to any cause related to or aggravated by the pregnancy, but not due to accidental or incidental causes. We will evaluate the follow outcome according the International Classification of Diseases–ICD-10: ICD-10 "XV" codes will be considered (Pregnancy, childbirth and the puerperium (O00-O99) to compose cases of maternal death, except for deaths after 42 days, "O96" and "O97"; and other ICD-10 chapters (A34, F53, M83.0, B20 to B24, D39.2, and E23.0) [68].

**iii) Child malnutrition.** *Study population*. The study will include data from the 100 Million Brazilian Cohort linked to SISVAN and SINASC. The study population will consist of children aged 0 to 5 years registered in the cohort baseline from 2004 to 2015. Anthropometric information from the last visit in the 2008 to 2017 period will be used.

*Definition of exposure*. Exposure in the studied population will consist of children who started and did not stop receiving the BFP before the last visit (2008 to 2017) to answer the objective of interest. Those not exposed will be the ones who have not received the benefit at any time until the date of the child's last visit.

*Outcome*. Nutritional status in children will be computed according to the WHO growth references and cutoff points for standardized height-for-age z-score (HAZ), weight-for-age z-score (WAZ), and weight-for-height z-score (WHZ) [69]. Anthropometric indices will be considered as continuous and categorized measures (Table 3).

## Statistical analysis

The effect of BFP on birth weight, preterm birth, maternal mortality, and child growth will be estimated based on propensity score-based methods (PS). The PS can be characterized as the conditional probability of receiving the treatment (to be a BFP beneficiary or not), given its observable characteristics [70]. These methods are different from the others in that they avoid multidimensionality and can be implemented using a control variable, which is the propensity score itself [46].

First, the missing data patterns will be evaluated for the variables considered in the calculation of the PS. Depending on these analyses, the PS calculation can be performed only with complete data or including the missing data as a category in each variable. The PS will be estimated using a logit model with baseline covariables related to receipt of BFP according Chart 1 (supplementary material).

The models will be weighted by the Inverse Probability Treatment Weighting (IPTW) and by the Kernel weights. Balancing will be performed before and after weighting to ensure that

the procedure used controlled for the available confounders. Finally, weighted Logistic Regressions and the Average Treatment Effect on Treated (ATT) will be calculated using non-linear and linear models, depending on the analyzed outcome [71, 72].

**Robustness analysis for propensity score-based methods.** As it is a dynamic cohort, analyses will be considered according to the treatment exposure time. Supplementary analyses will also be carried out with subpopulations with similar lengths of time since entering the cohort to balance the time until the outcome between recipients and controls. Also, analyses will be carried out for municipalities with a higher quality of information from vital statistics and according to the quantiles of coverage of the Family Health Strategy, region of residence and the decentralized (municipal) management index (IGD) of the BFP; and for subpopulations of maternal reproductive age (15 to 49 years or 10 to 49 years) [73, 74] and prenatal care follow-up.

## Ethical considerations

The Research Ethics Committee of the Institute of Collective Health, Federal University of Bahia (ICS-UFBA), approved the studies involved in this protocol under Opinion N° CAAE: 41695415.0.0000.5030 on May 30, 2017.

The linkage of the databases will be carried out in a secure environment, following a strict internal information security procedure to ensure data privacy and confidentiality [21]. A non-identified database will be used for the analyses, which can only be accessed by previously authorized researchers, and all steps after obtaining the data will be carried out following the CIDACS information security culture.

## Discussion

This study will use propensity score-based methods to assess the BFP effect on maternal and child health outcomes in a large sample of poor and impoverished Brazilian households. The BFP might be expected to result in positive effects in all conditions related to difficulties in accessing health, education, social assistance, employment, and income, thus, improving maternal and child health conditions. The study will follow internationally recognized guidelines for conducting and disseminating the results of impact assessment studies, providing transparency in conducting data analysis, and greater comparability of results [25, 75, 76].

Some limitations must be considered. Information systems can include missing data and lack of relevant information on potential outcome and confounding variables, such as more specialized access and quality of prenatal or postnatal care indicators, which could allow a better understanding of the nuances of the intervention (for example, distance to the clinic or ability and training of health professionals). We will not explore the results of the BFP concerning the amount of the transfers granted. BFP is a binary variable in our study, and nuances related to the amount received and poverty levels will not be explored in this first proposal.

On the other hand, the large-scale data set will allow us to investigate comprehensively and in subpopulations the effects of BFP on maternal and child outcomes. The use of these databases will allow us exploring rarer outcomes with a high level of statistical power. The databases used in this study have national coverage, low under-registration, and some have already documented reliability [59]. Thus, this study will provide a comprehensive and representative analysis of the poor and extremely poor Brazilian population and reinforce the adequacy of these bases for epidemiological investigations [59]. The availability of a cohort with socioeconomic information linked to maternal and child health data provides us with the possibility to assess the effect of the BFP on these outcomes, considering a wide range of explanatory and confounding variables, and enabling the use of methods based on propensity scores.

## Dissemination of knowledge

This evaluation of BFP will provide tools and evidence to program management focused on poverty reduction and reduction of adverse outcomes related to maternal and child health. We will disseminate the data in scientific journals, reports, and policy briefings targeting policy-makers and civil society.

## Supporting information

**S1 File.**
(DOCX)

## Author Contributions

**Conceptualization:** Ila Rocha Falcão, Rita de Cássia Ribeiro-Silva, Flávia Jôse Oliveira Alves, Naiá Ortelan, Natanael J. Silva, Rosemeire L. Fiaccone, Marcia Furquim de Almeida, Andrea J. F. Ferreira, Laura C. Rodrigues, Maria Yury Ichihara, Mauricio L. Barreto.

**Data curation:** Maria Yury Ichihara, Mauricio L. Barreto.

**Formal analysis:** Ila Rocha Falcão, Flávia Jôse Oliveira Alves, Naiá Ortelan, Natanael J. Silva, Rosemeire L. Fiaccone, Júlia M. Pescarini, Cinthia Soares Lisboa.

**Funding acquisition:** Laura C. Rodrigues, Mauricio L. Barreto.

**Investigation:** Ila Rocha Falcão, Flávia Jôse Oliveira Alves, Naiá Ortelan, Rosemeire L. Fiaccone, Marcia Furquim de Almeida, Júlia M. Pescarini, Cinthia Soares Lisboa, Andrea J. F. Ferreira, Laura C. Rodrigues, Maria Yury Ichihara, Mauricio L. Barreto.

**Methodology:** Ila Rocha Falcão, Rita de Cássia Ribeiro-Silva, Flávia Jôse Oliveira Alves, Naiá Ortelan, Natanael J. Silva, Rosemeire L. Fiaccone, Júlia M. Pescarini, Cinthia Soares Lisboa, Elzo Pereira Pinto Júnior, Enny S. Paixao, Camila Silveira Silva Teixeira, Aline dos Santos Rocha, Srinivasa Vittal Katikireddi, M. Sanni Ali, Ruth Dundas, Alastair Leyland.

**Project administration:** Rita de Cássia Ribeiro-Silva, Rosemeire L. Fiaccone, Laura C. Rodrigues, Maria Yury Ichihara, Mauricio L. Barreto.

**Resources:** Laura C. Rodrigues, Maria Yury Ichihara, Mauricio L. Barreto.

**Supervision:** Rita de Cássia Ribeiro-Silva, Rosemeire L. Fiaccone, Marcia Furquim de Almeida, Júlia M. Pescarini, Maria Yury Ichihara, Mauricio L. Barreto.

**Writing – original draft:** Ila Rocha Falcão, Rita de Cássia Ribeiro-Silva, Flávia Jôse Oliveira Alves, Naiá Ortelan, Natanael J. Silva.

**Writing – review & editing:** Ila Rocha Falcão, Rita de Cássia Ribeiro-Silva, Flávia Jôse Oliveira Alves, Naiá Ortelan, Natanael J. Silva, Rosemeire L. Fiaccone, Marcia Furquim de Almeida, Júlia M. Pescarini, Cinthia Soares Lisboa, Elzo Pereira Pinto Júnior, Enny S. Paixao, Andrea J. F. Ferreira, Camila Silveira Silva Teixeira, Aline dos Santos Rocha, Srinivasa Vittal Katikireddi, M. Sanni Ali, Ruth Dundas, Alastair Leyland, Laura C. Rodrigues, Maria Yury Ichihara, Mauricio L. Barreto.

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
