## [Decision Letter · Decision Letter 0]

4 Nov 2021

PONE-D-21-08640

Evaluating the impact of Bolsa Familia, Brazil’s conditional cash transfer programme, on maternal and child health: a study protocol

PLOS ONE

Dear Dr. Falcão,

Thank you for submitting your manuscript to PLOS ONE. After careful consideration, we feel that it has merit but does not fully meet PLOS ONE’s publication criteria as it currently stands. Therefore, we invite you to submit a revised version of the manuscript that addresses the points raised during the review process.

The present study protocol sits amid important research questions and production of quality evidence regarding a major social program and maternal and child outcomes. The rationale is also well stated. However, as consistently pointed out by the reviewers, there are several methodological aspects that need to be clarified, including steps for data management and availability, statistical procedures (also considering the huge sample size), and approaches for sensitivity analysis --all crucial for ensuring reproducibility. In addition, please note that more detail on the planned developments may be useful to highlight the need to register this study protocol, as opposed to bringing such information in the methodology section of derived original studies (for instance, by the same research group: https://journals.plos.org/plosmedicine/article?id=10.1371/journal.pmed.1003509).

We look forward to receiving your revised manuscript.

Kind regards,

Bárbara Hatzlhoffer Lourenço, Ph.D.

Academic Editor

PLOS ONE

Journal Requirements:

2. Please include your tables as part of your main manuscript and remove the individual files. Please note that supplementary tables (should remain/ be uploaded) as separate "supporting information" files"

4. Thank you for stating the following financial disclosure: "The funders had and will not have a role in study design, data collection and analysis, decision to publish, or preparation of the manuscript." 

6. We note that you have indicated that data from this study are available upon request. PLOS only allows data to be available upon request if there are legal or ethical restrictions on sharing data publicly. For more information on unacceptable data access restrictions, please see http://journals.plos.org/plosone/s/data-availability#loc-unacceptable-data-access-restrictions. 

7. Your abstract cannot contain citations. Please only include citations in the body text of the manuscript, and ensure that they remain in ascending numerical order on first mention.

8. Your ethics statement should only appear in the Methods section of your manuscript. If your ethics statement is written in any section besides the Methods, please delete it from any other section. 

9. Please upload a new copy of Figure 1 as the detail is not clear. Please follow the link for more information: " ext-link-type="uri" xlink:type="simple">https://blogs.plos.org/plos/2019/06/looking-good-tips-for-creating-your-plos-figures-graphics/"
" ext-link-type="uri" xlink:type="simple">https://blogs.plos.org/plos/2019/06/looking-good-tips-for-creating-your-plos-figures-graphics/"

10. Please include captions for your Supporting Information files at the end of your manuscript, and update any in-text citations to match accordingly. Please see our Supporting Information guidelines for more information: http://journals.plos.org/plosone/s/supporting-information. 

Reviewers' comments:

Reviewer's Responses to Questions

**Comments to the Author**

1. Does the manuscript provide a valid rationale for the proposed study, with clearly identified and justified research questions?

Reviewer #1: Yes

Reviewer #2: Yes

Reviewer #3: Yes

2. Is the protocol technically sound and planned in a manner that will lead to a meaningful outcome and allow testing the stated hypotheses?

Reviewer #1: Yes

Reviewer #2: Partly

Reviewer #3: Partly

3. Is the methodology feasible and described in sufficient detail to allow the work to be replicable?

Reviewer #1: No

Reviewer #2: No

Reviewer #3: Yes

4. Have the authors described where all data underlying the findings will be made available when the study is complete?

Reviewer #1: Yes

Reviewer #2: No

Reviewer #3: Yes

5. Is the manuscript presented in an intelligible fashion and written in standard English?

Reviewer #1: Yes

Reviewer #2: Yes

Reviewer #3: Yes

6. Review Comments to the Author

You may also provide optional suggestions and comments to authors that they might find helpful in planning their study.

Reviewer #1: Review:

Thank you for this informative paper on your proposed study protocol for exploring the impact of Brazil’s Bolsa Familia Program on maternal and child health outcomes. I appreciate the time you’ve put into describing your protocols and methods before undertaking a study, which is important for transparency in scientific research.

Suggested revisions:

1. In the first paragraph of the ‘Background’ section (lines 53-57), I’m not sure if this sentence refers to Brazil, lower-middle-income/higher-middle-income countries, globally. Please clarify.

2. In the first paragraph of the ‘Methods’ section, could you briefly elaborate on the original purpose of the 100 Million Brazilian Cohort survey.

3. In the second paragraph of the ‘Methods’ section, I find the sentence on lines 97-98 unclear. Can you briefly explain the two stages? Is this unique to your study? Or is this standard procedure for linking surveys to government data?

4. The second paragraph of the ‘Methods’ section is generally a bit disjointed with the CIDACS acronym being defined at the end, CIDACS-RL being mentioned, then linkages, then an explanation of CIDACS-RL. It could be improved for the ease of comprehension.

5. Line 105, it would be helpful to general readers to briefly explain the purpose of TIDieR-PHP reporting guidelines and why you used them.

6. Line 167, is definition b) not covered by definition a)? If so, then definition b) is redundant. If not, please rephrase b) for better clarity.

7. Line 180, do you want a ‘2)’ before “…stratified…”? Is it not a second definition?

8. You’ve mentioned Regression Discontinuity Design (RDD) in your keywords. I see no reference to RDD analyses in the main text. How do you plan to use RDD with your data? What questions do you hope to answer?

9. Rationale for stratification/sub-analyses of samples post-2011 is not clearly laid out in the text. It is briefly explained under “iii) child malnutrition - study population”. But post-2011 sub-analyses are suggested before that without explanation (LGA, SGA and prematurity). I figure this is due to changes in the BFP, but this is not clearly laid out in the text.

10. Table 1:

• In the text you refer to the Bolsa Familia Program, but in the table, it is the Family Grant Program. I suggest you use Bolsa Familia here for consistency.

• Under relevant variables from SINASC, it says “month that started prenatal after 2011”. Do you mean prenatal classes? Prenatal vitamins? Please check this.

• Are all the main variables you intend to use in your analyses listed in this table? It would be very important to know other health-related variable of the mothers (i.e. pre-existing diabetes or gestational diabetes; smoking; etc.) as these will be relevant confounders for prematurity, LGA and SGA analyses.

11. Table 3: There is an extra ‘e’ in front of “extremely preterm” in the outcomes column.

12. Figure 1: Resolution needs to be checked as it is barely readable at the moment.

For further consideration:

Your proposed rationale is reasonable, but have you considered how cash transfers that are conditional and preferentially paid to mothers may not increase purchasing power/ empowerment for all women. For example, the responsibility of getting children to school and to regular medical appointments for working mothers with partners may further entrench domestic/care work as women’s roles – potentially at the expense of paid employment, social networks, self-care, etc. Indeed, there appears to be a heterogeneous effect of CCT on women’s empowerment that may need further consideration in additional analyses:

De Brauw, A., Gilligan, D. O., Hoddinott, J., Roy, S. (2014). The impact of Bolsa Família on women’s decision-making power. World Development, 59, 487-504.

Reviewer #2: It is not clear to me that what the manuscript describes warrants a study protocol. The construction of the database itself has been published elsewhere by the same group. All aspects described in the section “Secondary objectives, study population, definition of exposure, and outcomes” are minor and would be well suited to methodology sections of different papers. The statistical methods are solid for natural experiment studies in public health. I firmly believe that the authors should expand the details of their methodological decisions and processes in a future submission and publish the product of this development as a supplemental file to their methodology.

Nevertheless, if the authors decide to proceed with the submission, some key points should be addressed.

Keeping in mind that reproducibility is one of the main pillars of study protocols at PLOS ONE, the authors should provide a comprehensive background of how the databases that compose the 100 Million Brazilian Cohort can be accessed for research purposes. If only governmental officials can access the data, the authors should consider another type of publication for this manuscript.

Given the 100 Million Brazilian complexity, the author should also expand on how they plan to address bias in all three outcomes.

A more thorough explanation should be provided concerning data cleaning decisions and the linkage process.

Minor points to be addressed:

Some aspects of the study population, exposure to PBF, and outcome should be standardized between sections. For example, picking either the “2004-2015” or “2004 to 2015” to declare year ranges.

In the logic model, “Linkage to the place of birth” is not a product but a process and does not belong to this logic model.

Reviewer #3: “Evaluating the Impact of Bolsa Familia, Brazil’s conditional cash transfer programme on, on maternal and child health: a study protocol,” submitted to PLOS ONE (PONE-08640)

The protocol outlines a substantive analysis using various large administrative health and social program databases from Brazil to the so-called 100 million Brazilian cohort. It represents an ambitious research agenda (likely leading to multiple papers) that has potential to improve understanding of the influences of PBF. The authors make clear that despite its enormous size and importance, careful empirical assessment of the effect of PBF, particularly on child and maternal outcomes, is sparse. They identify an appropriate set of outcomes based on available data and the literature. There are likely to have sufficient power for even very small impacts and rare events such as maternal mortality (making it important to judge not only statistical impact but size of impact). The large sample sizes and observational nature of the data make the research design, i.e., arguments for assessing causal impact and not just associations, crucial.

Main Comments:

1. One important reason there is not more evidence on PBF is the lack of a strong research design for assessing program impacts, as was done for example via randomization of Progresa in Mexico. Another challenge when examining impact at a national level, I believe, are the differences in program administration across municipalities played. The team proposes resolving identification of the causal effects of the intervention via propensity score matching techniques. This approach is preferable to simpler comparisons but still relies on key assumptions of non-confoundedness across treatment and comparison groups after matching. After controlling for the observed factors available, the assumption is that there are no unobserved differences in those taking up the treatment and those not taking it up. Central problem is that those who enter, despite observed characteristics, might be different – ie more likely to benefit or have unobserved wealth or something we do not observe. So, sign of bias is difficult to ascertain. If untreated are better off on other characteristics, for example, might be able to argue results are conservative or underestimates of beneficial program benefits. Unconfoundedness cannot be proven but the matching literature provides various approaches for assessing it in the articles cited and my expectation is many of these will be done in your analyses.

2. In practice, carrying out PS or other matching techniques on these large data sets will involve dozens if not hundreds of decisions regarding the specifics (on which variables, functional form, common support etc.) and possibly variations on those decisions to assess sensitivity. It could be useful to say a bit more about how this will be approached, including the specifics of the data for readers less familiar with it – for example specific variables/measures that are included or links to those descriptions or an appendix.

3. In their work (and related approaches), Imbens and coauthors develop other types of matching such as Nearest Neighbor (implemented in Stata using the command nnmatch). It may not be feasible with such large data sets to follow those approaches but a key aspect of them is allowing “exact” matching on certain types of characteristics. One that may be particularly important in this context is location – I noted mention of some subgroup analyses but think my suggestion here is a little different. Taking for example geographic location, to help ensure important elements like potential differences in municipality health systems are not leading to bias, a strategy of only matching treated cases with untreated in the same municipality can be used in the overall analysis. This permits an arguably better comparison than allowing geographic location, for example, to enter only via the combined propensity score.

4. Because PBF had an income cut-off, I did wonder whether there was any scope for an alternative approach to identification, related to regression discontinuity designs (RDD). I believe this could be done in conjunction with ps matching, but it would require availability of income measures (but these appear to be available). Or if not explicit, limiting comparison samples to those with incomes nearer to the cutoff, for example.

5. Administrative data match quality: I am unfamiliar with the various administrative data the study will use. It has been my experience in other settings, however, that combining administrative data across systems can be error ridden. To that end, greater support for the case that merging administrative records across the data sets is feasible and result in high quality (and high %) matches would strengthen confidence in the research design and the ultimate findings. Differences in quality of administrative match across the different data sets may influence findings in the three domains differently. It was unclear to me what the “similarity index” (page 3) approach was, but I presume on subsets of information (eg birth date, gender, location but not quite exact name spelling). A clear distinction in the final papers between the administrative matching across data sets and the ps matching procedures needs to be made. Characteristics of those matched and those not matched could shed light on potential biases.

6. Are there any statistical considerations relevant to having particularly large sample sizes?

7. It was unclear to me whether length of exposure (beyond pregnancy periods) for outcomes would be considered, but I may have missed this.

7. PLOS authors have the option to publish the peer review history of their article (what does this mean?). If published, this will include your full peer review and any attached files.

Reviewer #1: No

Reviewer #2: No

Reviewer #3: No

---

## [Author Response · Author response to Decision Letter 0]

8 Feb 2022

To the Editor and reviewers,

First, we would like to express our appreciation for your consideration of our manuscript, especially in light of the crisis the world is currently experiencing. We are very grateful for the pertinent criticism offered, and believe the incorporation of the reviewer’s suggestions will greatly contribute to the quality of our publication. Please find our point-by-point responses below to the criticism raised by each reviewer: 

Editor:

The present study protocol sits amid important research questions and production of quality evidence regarding a major social program and maternal and child outcomes. The rationale is also well stated. 

However, as consistently pointed out by the reviewers, there are several methodological aspects that need to be clarified, including steps for data management and availability, statistical procedures (also considering the huge sample size), and approaches for sensitivity analysis --all crucial for ensuring reproducibility. 

In addition, please note that more detail on the planned developments may be useful to highlight the need to register this study protocol, as opposed to bringing such information in the methodology section of derived original studies (for instance, by the same research group: https://journals.plos.org/plosmedicine/article?id=10.1371/journal.pmed.1003509).

- Please find our point-by-point responses below to the criticism raised by each reviewer.

Journal Requirements:

2. Please include your tables as part of your main manuscript and remove the individual files. Please note that supplementary tables (should remain/ be uploaded) as separate "supporting information" files"

- We apologize for this oversight, and we have included the tables and figures in manuscript.

- We apologize for this oversight, and we have changed this information.

4. Thank you for stating the following financial disclosure: "The funders had and will not have a role in study design, data collection and analysis, decision to publish, or preparation of the manuscript." 

- We appreciate your comment. We have included the financial disclosure (ls. 362-64).

- Thank you for your comment. We have revised and separated in "Funding information" the individual and institutional fundings.

- We have added the statements to the cover letter as requested.

- This study protocol will use linked data from the 100 Million Brazilian Cohort, coordinated and housed by the Center for Data and Knowledge Integration for Health (CIDACS). In accordance with the policy of CIDACS and the Ministry of Health and Ministry of Citizenship (database providers), restrictions on the availability of this data apply. Currently, only national and international researchers who collaborate with CIDACS and authorized staff from government agencies can access de-identified or anonymized linked data. Any person who wishes to receive authorization must: (i) be affiliated to CIDACS or be accepted as collaborators; (ii) present a detailed research project together with approval by an appropriate Brazilian institutional research ethical committee; (iii) provide a clear data plan restricted to the objectives of the proposed study and a summary of the analyses plan intended to guide the linkage and or data extraction of the relevant set of records and variables; (iv) sign terms of responsibility regarding the access and use of data; and (v) perform the analyses of datasets provided using the CIDACS data environment, a safe and secure infrastructure that provides remote access to de-identified or anonymized datasets and analysis tools. For more information, please visit the CIDACS website [https://cidacs.bahia.fiocruz.br/] or contact us via email [cidacs@fiocruz.br]. 

- We updated and included the Data Availability in the cover letter.

6. We note that you have indicated that data from this study are available upon request. PLOS only allows data to be available upon request if there are legal or ethical restrictions on sharing data publicly. For more information on unacceptable data access restrictions, please see http://journals.plos.org/plosone/s/data-availability#loc-unacceptable-data-access-restrictions. 

- This study protocol will use linked data from the 100 Million Brazilian Cohort, coordinated and housed by the Center for Data and Knowledge Integration for Health (CIDACS). The cohort is composed of admistrative databases, which the owners, the Brazil’s Ministry of Health and Ministry of Citizenship, have given CIDACS the custody and authorization to conduct research, with the guarantee that all data processing takes place in a safe and private environment. Therefore, importante restrictions apply to the availability of this data.

Currently, only national and international researchers who collaborate with CIDACS and authorized staff from government agencies can access de-identified or anonymized linked data. Any person who wishes to receive authorization must: (i) be affiliated to CIDACS or be accepted as collaborators; (ii) present a detailed research project together with approval by an appropriate Brazilian institutional research ethical committee; (iii) provide a clear data plan restricted to the objectives of the proposed study and a summary of the analyses plan intended to guide the linkage and or data extraction of the relevant set of records and variables; (iv) sign terms of responsibility regarding the access and use of data; and (v) perform the analyses of datasets provided using the CIDACS data environment, a safe and secure infrastructure that provides remote access to de-identified or anonymized datasets and analysis tools. For more information, please visit the CIDACS website [https://cidacs.bahia.fiocruz.br/] or contact us via email [cidacs@fiocruz.br]. 

This study protocol has already received the approval and authorization from CIDACS. In addition, the Research Ethics Committee of the Institute of Collective Health, Federal University of Bahia (ICS-UFBA), approved the study under Opinion N° CAAE: 41695415.0.0000.5030. All steps subsequent to obtaining the data will be carried out following the CIDACS information security protocols.

- In line with the previous answer, it is important to note that there are restrictions that apply to the availability of these data, which are licensed for use in the present study only and therefore are not publicly available. We updated the Data Availability statement as requested avove. Recently, an article was published that gives more details about the databases used to build the 100 Million Brazilian Cohort (doi: 10.1093/ije/dyab213)

7. Your abstract cannot contain citations. Please only include citations in the body text of the manuscript, and ensure that they remain in ascending numerical order on first mention.

- We apologize for this oversight, and we have removed this citation from the abstract.

8. Your ethics statement should only appear in the Methods section of your manuscript. If your ethics statement is written in any section besides the Methods, please delete it from any other section. 

- Thank you for your comment. Our ethics statement only appears in the Methods section and in Declarations, according Plos One requirements.

9. Please upload a new copy of Figure 1 as the detail is not clear. Please follow the link for more information: https://blogs.plos.org/plos/2019/06/looking-good-tips-for-creating-your-plos-figures-graphics/" https://blogs.plos.org/plos/2019/06/looking-good-tips-for-creating-your-plos-figures-graphics/"

- We added a new copy of Figure 1 into the manuscript.

10. Please include captions for your Supporting Information files at the end of your manuscript, and update any in-text citations to match accordingly. Please see our Supporting Information guidelines for more information: http://journals.plos.org/plosone/s/supporting-information. 

- We included captions for our Supporting Information files at the end of our manuscript.

Reviewers' comments:

Reviewer's Responses to Questions

Comments to the Author

1. Does the manuscript provide a valid rationale for the proposed study, with clearly identified and justified research questions?

Reviewer #1: Yes

Reviewer #2: Yes

Reviewer #3: Yes

- We thank the reviewers for his/her careful reading of our text. 

2. Is the protocol technically sound and planned in a manner that will lead to a meaningful outcome and allow testing the stated hypotheses?

Reviewer #1: Yes

Reviewer #2: Partly

Reviewer #3: Partly

- We appreciate your important comment. We describe all the objectives of the studies to be carried out. In addition, we present a logic model to describe the hypothetical mechanisms by which BFP may affect maternal and infant outcomes (Figure 1).

3. Is the methodology feasible and described in sufficient detail to allow the work to be replicable?

Reviewer #1: No

Reviewer #2: No

Reviewer #3: Yes

- Working with population databases gives us many possibilities to assess specific subpopulations and explore hypotheses, but it has some limitations. We will work with a database of people eligible for social programs, which can be up to 90% beneficiaries and 10% non-beneficiaries. Thus, we do not have a sample, we cannot previously define a number for the control group, and we need appropriate methods. Thus, we believe that matching may not be the best way, but the methods based on Propensity Score (weighting), could be applied.

4. Have the authors described where all data underlying the findings will be made available when the study is complete?

Reviewer #1: Yes

Reviewer #2: No

Reviewer #3: Yes

- As mentioned earlier, it is important to note that there are restrictions that apply to availability of such data, which are licensed for use only in the present study and therefore not able to make available publicly. Recently, two articles were published that provide more details on the databases used to build the 100 Million Brazilian Cohort (doi: 10.1093/ije/dyab213) and CIDACS Birth Cohort (DOI: 10.1093/ije/dyaa255)

5. Is the manuscript presented in an intelligible fashion and written in standard English?

Reviewer #1: Yes

Reviewer #2: Yes

Reviewer #3: Yes

- We thank the reviewers. 

6. Review Comments to the Author

You may also provide optional suggestions and comments to authors that they might find helpful in planning their study.

Reviewer #1: Review:

Thank you for this informative paper on your proposed study protocol for exploring the impact of Brazil’s Bolsa Familia Program on maternal and child health outcomes. I appreciate the time you’ve put into describing your protocols and methods before undertaking a study, which is important for transparency in scientific research.

Suggested revisions:

1. In the first paragraph of the ‘Background’ section (lines 53-57), I’m not sure if this sentence refers to Brazil, lower-middle-income/higher-middle-income countries, globally. Please clarify.

- We appreciate the reviewer’s comments. The sentence refers to low and-middle-income countries. The sentence in question incorporated this notion: “…low birth weight, and child malnutrition, especially among the low and -middle-income countries (LMIC), hinder the achievement of the Sustainable Development Goals (SDGs) [4-8].”

2. In the first paragraph of the ‘Methods’ section, could you briefly elaborate on the original purpose of the 100 Million Brazilian Cohort survey.

- We agree with this suggestion. We have added appropriate citations in the text: “The main objective of the 100 million Brazilian cohort is to enable the study of the social determinants and the effects of social policies and programs on the different aspects of health in Brazil (https://doi.org/10.1093/ije/dyab213). ”

3. In the second paragraph of the ‘Methods’ section, I find the sentence on lines 97-98 unclear. Can you briefly explain the two stages? Is this unique to your study? Or is this standard procedure for linking surveys to government data?

- The linkage procedures are common for 100 Million Brazilian Cohort studies. We have included other recent publication and the mention: “We will use CIDACS Record Linkage (CIDACS-RL) to link the databases (https://doi.org/10.1186/s12911-020-01285-w). The linkage procedures consist of two stages. The first will be a deterministic linkage, and the second will be based on the similarity index. More detailed information can be consulted in previous publications (https://doi.org/10.1186/s12911-020-01192-0) (https://doi.org/10.3389/fphar.2019.00984).”

4. The second paragraph of the ‘Methods’ section is generally a bit disjointed with the CIDACS acronym being defined at the end, CIDACS-RL being mentioned, then linkages, then an explanation of CIDACS-RL. It could be improved for the ease of comprehension.

- We apologize for this oversight and have added an explanation in line 100.

5. Line 105, it would be helpful to general readers to briefly explain the purpose of TIDieR-PHP reporting guidelines and why you used them. 

- We have added an explanation in ls. 113-4. The checklist consists of 9 items and helps researchers to describe the characteristics of population health and policy interventions.

6. Line 167, is definition b) not covered by definition a)? If so, then definition b) is redundant. If not, please rephrase b) for better clarity.

- Thank you. We have removed the definition b).

7. Line 180, do you want a ‘2)’ before “…stratified…”? Is it not a second definition?

- We have added the “2” before “…stratified…”.

8. You’ve mentioned Regression Discontinuity Design (RDD) in your keywords. I see no reference to RDD analyses in the main text. How do you plan to use RDD with your data? What questions do you hope to answer?

- We have removed RDD among the keywords.

9. Rationale for stratification/sub-analyses of samples post-2011 is not clearly laid out in the text. It is briefly explained under “iii) child malnutrition - study population”. But post-2011 sub-analyses are suggested before that without explanation (LGA, SGA and prematurity). I figure this is due to changes in the BFP, but this is not clearly laid out in the text.

- We agree with your comment. After discussion with the team, we came to the conclusion that it will not be possible, considering the impossibility of working with the variable income. We included a mention about SGA and LGA (ls 161-2). For the "SGA" and "LGA" outcomes, it is only possible to assess after 2011, as we do not have any information on gestational age before that.

10. Table 1:

• In the text you refer to the Bolsa Familia Program, but in the table, it is the Family Grant Program. I suggest you use Bolsa Familia here for consistency.

- We apologize for this mistake. We changed for Bolsa Familia Program.

• Under relevant variables from SINASC, it says “month that started prenatal after 2011”. Do you mean prenatal classes? Prenatal vitamins? Please check this.

- This refers to the month of pregnancy in which the woman started the prenatal consultations. We only have the variables number of consultations (2004 to 2015) and month in which prenatal care started (2011 to 2015). To make the information clearer, we have included the information in the table: “Some variables such as the month in which the woman started prenatal care and gestational age (as a continuous variable) are only available for the period from 2011 to 2015. ”

• Are all the main variables you intend to use in your analyses listed in this table? It would be very important to know other health-related variable of the mothers (i.e. pre-existing diabetes or gestational diabetes; smoking; etc.) as these will be relevant confounders for prematurity, LGA and SGA analyses.

- We have included all possible variables to be evaluated. Unfortunately, the database does not have important variables such as, smoking, physical activity or alcohol use.

11. Table 3: There is an extra ‘e’ in front of “extremely preterm” in the outcomes column.

- We apologize for this mistake. We changed this.

12. Figure 1: Resolution needs to be checked as it is barely readable at the moment.

- We have included a figure with better resolution.

For further consideration:

Your proposed rationale is reasonable, but have you considered how cash transfers that are conditional and preferentially paid to mothers may not increase purchasing power/ empowerment for all women. For example, the responsibility of getting children to school and to regular medical appointments for working mothers with partners may further entrench domestic/care work as women’s roles – potentially at the expense of paid employment, social networks, self-care, etc. Indeed, there appears to be a heterogeneous effect of CCT on women’s empowerment that may need further consideration in additional analyses:

De Brauw, A., Gilligan, D. O., Hoddinott, J., Roy, S. (2014). The impact of Bolsa Família on women’s decision-making power. World Development, 59, 487-504.

- We appreciate and agree with yourcomment. Indeed, the effect of Bolsa Família on female empowerment still needs further study due to the heterogeneous results, as mentioned. 

Reviewer #2: It is not clear to me that what the manuscript describes warrants a study protocol. The construction of the database itself has been published elsewhere by the same group. All aspects described in the section “Secondary objectives, study population, definition of exposure, and outcomes” are minor and would be well suited to methodology sections of different papers. The statistical methods are solid for natural experiment studies in public health. I firmly believe that the authors should expand the details of their methodological decisions and processes in a future submission and publish the product of this development as a supplemental file to their methodology.

Nevertheless, if the authors decide to proceed with the submission, some key points should be addressed.

- We appreciate your suggestion. We strongly believe that designing a research protocol and submittingg it for critical peer review and publication before conducting studies, especially intervention studies, are important to describe the methods in suficiente detail and prevent undisclosed flexibility in the experimental procedures and analysis. We believe that it is a good practice for research, providing more quality and clarity to future findings. For these reasons we have decided to proceed with submission.

Keeping in mind that reproducibility is one of the main pillars of study protocols at PLOS ONE, the authors should provide a comprehensive background of how the databases that compose the 100 Million Brazilian Cohort can be accessed for research purposes. If only governmental officials can access the data, the authors should consider another type of publication for this manuscript.

Given the 100 Million Brazilian complexity, the author should also expand on how they plan to address bias in all three outcomes.

A more thorough explanation should be provided concerning data cleaning decisions and the linkage process.

- This study protocol will use linked data from the 100 Million Brazilian Cohort, coordinated and housed by the Center for Data and Knowledge Integration for Health (CIDACS). The cohort is composed of admistrative databases, which the owners, the Brazil’s Ministry of Health and Ministry of Citizenship, have given CIDACS the custody and authorization to conduct research, with the guarantee that all data processing takes place in a safe and private environment. Therefore, importante restrictions apply to the availability of this data.

Currently, only national and international researchers who collaborate with CIDACS and authorized staff from government agencies can access de-identified or anonymized linked data. Any person who wishes to receive authorization must: (i) be affiliated to CIDACS or be accepted as collaborators; (ii) present a detailed research project together with approval by an appropriate Brazilian institutional research ethical committee; (iii) provide a clear data plan restricted to the objectives of the proposed study and a summary of the analyses plan intended to guide the linkage and or data extraction of the relevant set of records and variables; (iv) sign terms of responsibility regarding the access and use of data; and (v) perform the analyses of datasets provided using the CIDACS data environment, a safe and secure infrastructure that provides remote access to de-identified or anonymized datasets and analysis tools. For more information, please visit the CIDACS website [https://cidacs.bahia.fiocruz.br/] or contact us via email [cidacs@fiocruz.br]. 

We have been working with different Linked databases. Recently, two articles were published that provide more details on the databases used to build the 100 Million Brazilian Cohort (doi: 10.1093/ije/dyab213) and CIDACS Birth Cohort (DOI: 10.1093/ije/dyaa255). 

As mentioned earlier, the linkage procedures are common for 100 Million Cohort studies. We have included a recent publications and the mention: “We will use CIDACS Record Linkage (CIDACS-RL) to link the databases (https://doi.org/10.1186/s12911-020-01285-w). The linkage procedures consist of two stages. The first will be a deterministic linkage, and the second will be based on the similarity index. Others detailed information can be consulted in previous publications (https://doi.org/10.3389/fphar.2019.00984 and https://doi.org/10.1186/s12911-020-01192-0).” It is important to note that this protocol covers many studies and each specific objective involves a different database. Analyzes and cleaning procedures are specific to each database covered in this protocol. We believe that the mention of methodological studies, already published, can provide more detailed information about common procedures (linkage method) and general characteristics of the cohort.

Minor points to be addressed:

Some aspects of the study population, exposure to PBF, and outcome should be standardized between sections. For example, picking either the “2004-2015” or “2004 to 2015” to declare year ranges.

- We have standardized the ranges as suggested.

In the logic model, “Linkage to the place of birth” is not a product but a process and does not belong to this logic model.

- We agree with your comment, and we withdraw the term from the logic model.

Reviewer #3: “Evaluating the Impact of Bolsa Familia, Brazil’s conditional cash transfer programme on, on maternal and child health: a study protocol,” submitted to PLOS ONE (PONE-08640)

The protocol outlines a substantive analysis using various large administrative health and social program databases from Brazil to the so-called 100 million Brazilian cohort. It represents an ambitious research agenda (likely leading to multiple papers) that has potential to improve understanding of the influences of PBF. The authors make clear that despite its enormous size and importance, careful empirical assessment of the effect of PBF, particularly on child and maternal outcomes, is sparse. They identify an appropriate set of outcomes based on available data and the literature. There are likely to have sufficient power for even very small impacts and rare events such as maternal mortality (making it important to judge not only statistical impact but size of impact). The large sample sizes and observational nature of the data make the research design, i.e., arguments for assessing causal impact and not just associations, crucial.

Main Comments:

1. One important reason there is not more evidence on PBF is the lack of a strong research design for assessing program impacts, as was done for example via randomization of Progresa in Mexico. Another challenge when examining impact at a national level, I believe, are the differences in program administration across municipalities played. The team proposes resolving identification of the causal effects of the intervention via propensity score matching techniques. This approach is preferable to simpler comparisons but still relies on key assumptions of non-confoundedness across treatment and comparison groups after matching. After controlling for the observed factors available, the assumption is that there are no unobserved differences in those taking up the treatment and those not taking it up. Central problem is that those who enter, despite observed characteristics, might be different – ie more likely to benefit or have unobserved wealth or something we do not observe. So, sign of bias is difficult to ascertain. If untreated are better off on other characteristics, for example, might be able to argue results are conservative or underestimates of beneficial program benefits. Unconfoundedness cannot be proven but the matching literature provides various approaches for assessing it in the articles cited and my expectation is many of these will be done in your analyses.

- We appreciate your careful reading and we are very grateful for the pertinent criticism offered. Firstly we agree that results probably will be conservative or underestimate of beneficial program benefits. Second, we propose the effec evaluation via methods based on propensity score (weighting). Working with population databases gives us many possibilities to assess specific subpopulations and explore hypotheses, but it has some limitations. We will work with a database of people eligible for social programs, which can be up to 90% beneficiaries and 10% non-beneficiaries. Thus, we do not have a sample, we cannot previously define a number for the control group, and we need appropriate methods. In this sense, we believe that methods based on Propensity Score (weighting), could be applied. If it is known that these are procedures related to quasi-experimental methods, we are careful with the use of the term “causal effect”. We will use a large dataset with many confounding covariates, but unfortunately we cannot guarante unconfoundedness.

We will perform balancing after weighting to ensure that the procedure used controlled for available confounders. We have included the mention in the text: “Balancing will be performed before and after weighting to ensure that the procedure used controlled for the available confounders.” We will also do analyzes according to subpopulations (Robustness analysis for propensity score-based methods section).

2. In practice, carrying out PS or other matching techniques on these large data sets will involve dozens if not hundreds of decisions regarding the specifics (on which variables, functional form, common support etc.) and possibly variations on those decisions to assess sensitivity. It could be useful to say a bit more about how this will be approached, including the specifics of the data for readers less familiar with it – for example specific variables/measures that are included or links to those descriptions or an appendix.

- We appreciate your suggestion and have included a chart (chart 1) as supplementary material.

3. In their work (and related approaches), Imbens and coauthors develop other types of matching such as Nearest Neighbor (implemented in Stata using the command nnmatch). It may not be feasible with such large data sets to follow those approaches but a key aspect of them is allowing “exact” matching on certain types of characteristics. One that may be particularly important in this context is location – I noted mention of some subgroup analyses but think my suggestion here is a little different. Taking for example geographic location, to help ensure important elements like potential differences in municipality health systems are not leading to bias, a strategy of only matching treated cases with untreated in the same municipality can be used in the overall analysis. This permits an arguably better comparison than allowing geographic location, for example, to enter only via the combined propensity score.

- We appreciate your pertinent criticism. As mentioned earlier, we propose the effec evaluation via methods based on propensity score (weighting). We will work with a database of people eligible for social programs, which can be up to 90% beneficiaries and 10% non-beneficiaries. Thus, we do not have a sample, we cannot previously define a number for the control group, and we need appropriate methods. In this sense, we believe that methods based on Propensity Score (weighting), could be applied. We are still not sure if it would be possible to include the location in the propensity score calculation of with a level of disaggregation greater than the Region. This depends on the cases according to state or municipality. We intend to perform analysis by location subgroup (region, state...) and we can do it because we are using weighting (not matching), unless you have more than one region without cases.

4. Because PBF had an income cut-off, I did wonder whether there was any scope for an alternative approach to identification, related to regression discontinuity designs (RDD). I believe this could be done in conjunction with ps matching, but it would require availability of income measures (but these appear to be available). Or if not explicit, limiting comparison samples to those with incomes nearer to the cutoff, for example.

- We appreciate your comment. Due to changes to the income variable in the Cadúnico form, we cannot use this variable for the period from 2004 to 2015 (excessive missings and zeros). Thus, unfortunately, we will not be able to explore the RDD due to the impossibility of working with the income variable.

5. Administrative data match quality: I am unfamiliar with the various administrative data the study will use. It has been my experience in other settings, however, that combining administrative data across systems can be error ridden. To that end, greater support for the case that merging administrative records across the data sets is feasible and result in high quality (and high %) matches would strengthen confidence in the research design and the ultimate findings. Differences in quality of administrative match across the different data sets may influence findings in the three domains differently. It was unclear to me what the “similarity index” (page 3) approach was, but I presume on subsets of information (eg birth date, gender, location but not quite exact name spelling). A clear distinction in the final papers between the administrative matching across data sets and the ps matching procedures needs to be made. Characteristics of those matched and those not matched could shed light on potential biases.

- We have been working with different linked databases. Recently, two articles were published that provide more details on the databases used to build the 100 Million Brazilian Cohort (doi: 10.1093/ije/dyab213) and CIDACS Birth Cohort (DOI: 10.1093/ije/dyaa255). As mentioned earlier, the linkage procedures are common for 100 milion Cohort studies. We have included a recent publications and the mention: “We will use CIDACS Record Linkage (CIDACS-RL) to link the databases (https://doi.org/10.1186/s12911-020-01285-w). The linkage procedures are common for 100 milion Cohort studies and consist of two stages. The first will be a deterministic linkage, and the second will be based on the similarity index. Others detailed information can be consulted in previous publications (https://doi.org/10.3389/fphar.2019.00984 and https://doi.org/10.1186/s12911-020-01192-0). It is important to note that this protocol covers many studies and each specific objective involves a database. Analyzes are specific to each database covered in this protocol. We believe that the mention of methodological studies, already published, can provide more detailed information about common procedures (linkage method) and general characteristics of the cohort.

6. Are there any statistical considerations relevant to having particularly large sample sizes?

- Large sample sizes provide comprehensive data to conduct analyses on subgroups of interest while maintaining sufficient power to gain insights into the direction and size of the effects. It is important to highlight that larger samples provide great opportunities for empirical research, but also may lead to equivocal interpretations due to the detection of statistical significance. Below we indicate some publications.

Gelman A. P values and statistical practice. Epidemiology (Cambridge, Mass). 2013;24(1):69-72.

Siontis GCM, Ioannidis JPA. Risk factors and interventions with statistically significant tiny effects. International journal of epidemiology. 2011;40(5):1292-1307.

7. It was unclear to me whether length of exposure (beyond pregnancy periods) for outcomes would be considered, but I may have missed this.

- We apologize for the confusion of definitions. We created a chart to make these definitions clearer (Chart 1 in supplement).

7. PLOS authors have the option to publish the peer review history of their article (what does this mean?). If published, this will include your full peer review and any attached files.

Do you want your identity to be public for this peer review? For information about this choice, including consent withdrawal, please see our Privacy Policy.

Reviewer #1: No

Reviewer #2: No

Reviewer #3: No

---

## [Decision Letter · Decision Letter 1]

14 Mar 2022

PONE-D-21-08640R1Evaluating the impact of Bolsa Familia, Brazil’s conditional cash transfer programme, on maternal and child health: a study protocolPLOS ONE

Dear Dr. Falcão,

Thank you for submitting your manuscript to PLOS ONE. After careful consideration, we feel that it has merit but does not fully meet PLOS ONE’s publication criteria as it currently stands. Therefore, we invite you to submit a revised version of the manuscript that addresses the points raised during the review process.

We look forward to receiving your revised manuscript.

Kind regards,

Bárbara Hatzlhoffer Lourenço, Ph.D.

Academic Editor

PLOS ONE

Journal Requirements:

Reviewers' comments:

Reviewer's Responses to Questions

**Comments to the Author**

1. Does the manuscript provide a valid rationale for the proposed study, with clearly identified and justified research questions?

Reviewer #1: Yes

Reviewer #2: Yes

Reviewer #3: Yes

2. Is the protocol technically sound and planned in a manner that will lead to a meaningful outcome and allow testing the stated hypotheses?

Reviewer #1: Yes

Reviewer #2: Yes

Reviewer #3: Yes

3. Is the methodology feasible and described in sufficient detail to allow the work to be replicable?

Reviewer #1: Yes

Reviewer #2: Yes

Reviewer #3: Yes

4. Have the authors described where all data underlying the findings will be made available when the study is complete?

Reviewer #1: Yes

Reviewer #2: Yes

Reviewer #3: Yes

5. Is the manuscript presented in an intelligible fashion and written in standard English?

Reviewer #1: Yes

Reviewer #2: Yes

Reviewer #3: Yes

6. Review Comments to the Author

You may also provide optional suggestions and comments to authors that they might find helpful in planning their study.

Reviewer #1: Overall, the manuscript is much more clear. The authors have made important improvements. There are, however, a few remaining issues to be addressed:

1- Please check your acronyms. Sometimes you use PBF, and other times BFP.

2- Line 158-159: Why drop babies of less than 500g or born before 22 gestational weeks? I know they may be less likely to survive but could extreme prematurity and/or very low birthweight not be related to the socioeconomic standing of the mother? Is this not conditioning on the outcome? I understand the rationale for multiple births or congenital abnormalities as they are unlikely to be related to SES but related to birthweight and prematurity, but excluding the babies based on their likelihood of survival does not make sense to me.

3- On line 241, do you mean the parent's education or the child's?

4- Line 270: You say "quasi-experimental approaches". You should explain how your approaches are quasi-experimental (i.e. exogenous changes with the BF program and subsequent outcomes, or regression discontinuity designs around the cut points for BF eligibility). I'm not sure I would use this characterization here based on how your have described your proposed analyses.

Reviewer #2: The authors addressed all the points raised. I believe that this work have the potential to improve methodological rigor in future researches exploring maternal and child health outcomes

Reviewer #3: Thank you for the clarifications and revisions. In finalizing, I suggest you consider reconsidering words like "effect" and "impact" if you wish to be more careful around causal language (see you response to my first comment). In chart 1, column 1 it often says "access" but I believe you mean "assess"

7. PLOS authors have the option to publish the peer review history of their article (what does this mean?). If published, this will include your full peer review and any attached files.

Reviewer #1: No

Reviewer #2: No

Reviewer #3: **Yes: **John A. Maluccio

---

## [Author Response · Author response to Decision Letter 1]

20 Apr 2022

To the Editor and reviewers,

First, we would like to express our appreciation for your consideration of our manuscript. We are very grateful for the pertinent criticism offered, and believe the incorporation of the reviewer’s suggestions will greatly contribute to the quality of our publication. Please find our point-by-point responses below to the criticism raised by each reviewer:

Journal Requirements:

- We appreciate the suggestion. We have carefully reviewed our list of references. We did not have articles with retractions. We included newer government decrees, but kept those in force in the period defined by the study.

Reviewers' comments:

Reviewer's Responses to Questions

Comments to the Author

1. Does the manuscript provide a valid rationale for the proposed study, with clearly identified and justified research questions?

Reviewer #1: Yes

Reviewer #2: Yes

Reviewer #3: Yes

2. Is the protocol technically sound and planned in a manner that will lead to a meaningful outcome and allow testing the stated hypotheses?

Reviewer #1: Yes

Reviewer #2: Yes

Reviewer #3: Yes

3. Is the methodology feasible and described in sufficient detail to allow the work to be replicable?

Reviewer #1: Yes

Reviewer #2: Yes

Reviewer #3: Yes

4. Have the authors described where all data underlying the findings will be made available when the study is complete?

Reviewer #1: Yes

Reviewer #2: Yes

Reviewer #3: Yes

5. Is the manuscript presented in an intelligible fashion and written in standard English?

Reviewer #1: Yes

Reviewer #2: Yes

Reviewer #3: Yes

6. Review Comments to the Author

You may also provide optional suggestions and comments to authors that they might find helpful in planning their study.

Reviewer #1: Overall, the manuscript is much more clear. The authors have made important improvements. There are, however, a few remaining issues to be addressed:

1- Please check your acronyms. Sometimes you use PBF, and other times BFP.

- We appreciate your careful review. We revised the text and changed the acronym that was present in reference 15, line 449.

2- Line 158-159: Why drop babies of less than 500g or born before 22 gestational weeks? I know they may be less likely to survive but could extreme prematurity and/or very low birthweight not be related to the socioeconomic standing of the mother? Is this not conditioning on the outcome? I understand the rationale for multiple births or congenital abnormalities as they are unlikely to be related to SES but related to birthweight and prematurity, but excluding the babies based on their likelihood of survival does not make sense to me.

- We appreciate your comment. We consider that survival may be related to SES and better access to a specialized delivery service, we used this feasibility criterion (500g and 22sg). As pointed out, women with better SES could be more likely to have babies with 300g or 20sg (criteria used in developed countries), but we must emphasize that we are working with populations of socioeconomically vulnerable women. In a way, we agree with your comment and removed the mention of the criterion that will be applied, keeping only the phrase “Fetal viability criteria can be applied [62-65]. "

3- On line 241, do you mean the parent's education or the child's?

- We appreciate the comment. We apologize, but we could not find this passage in the text. When we refer to education in the text, we are usually referring to maternal schooling. Except in some passages of the text (when citing the BFP conditionalities), we refer to school attendance by adolescents (17).

4- Line 270: You say "quasi-experimental approaches". You should explain how your approaches are quasi-experimental (i.e. exogenous changes with the BF program and subsequent outcomes, or regression discontinuity designs around the cut points for BF eligibility). I'm not sure I would use this characterization here based on how your have described your proposed analyses.

- We appreciate your comment and agree that the term used was not so appropriate. We rewrite the sentence to “This study will use propensity score-based methods to assess the BFP effect on maternal and child health outcomes in a large sample of poor and impoverished Brazilian households.”

Reviewer #2: The authors addressed all the points raised. I believe that this work have the potential to improve methodological rigor in future researches exploring maternal and child health outcomes

- We appreciate your careful review.

Reviewer #3: Thank you for the clarifications and revisions. In finalizing, I suggest you consider reconsidering words like "effect" and "impact" if you wish to be more careful around causal language (see you response to my first comment). In chart 1, column 1 it often says "access" but I believe you mean "assess"

- We appreciate your careful review. We consider maintaining the term “effect”. We consider this to be justified by the limitations of each study generated from this protocol. We know that we cannot sustain the “causal effect” and that the effect of the Bolsa Família found will probably be associative, due to the characteristics of the program and the proposed analysis (adequate to the particularities of the program and the database).

- We have modified the terms in Chart 1 to “To evaluate”.

7. PLOS authors have the option to publish the peer review history of their article (what does this mean?). If published, this will include your full peer review and any attached files.

Do you want your identity to be public for this peer review? For information about this choice, including consent withdrawal, please see our Privacy Policy.

Reviewer #1: No

Reviewer #2: No

Reviewer #3: Yes: John A. Maluccio

- We appreciate the suggestion. The figure was generated from the PACE digital diagnostic tool and attached to this submission.

---

## [Editor Report · Decision Letter 2]

4 May 2022

Evaluating the effect of Bolsa Familia, Brazil’s conditional cash transfer programme, on maternal and child health: a study protocol

PONE-D-21-08640R2

Dear Dr. Falcão,

We’re pleased to inform you that your manuscript has been judged scientifically suitable for publication and will be formally accepted for publication once it meets all outstanding technical requirements.

Kind regards,

Bárbara Hatzlhoffer Lourenço, Ph.D.

Academic Editor

PLOS ONE
---

## [Editor Report · Acceptance letter]

12 May 2022

PONE-D-21-08640R2 

Evaluating the effect of Bolsa Familia, Brazil’s conditional cash transfer programme, on maternal and child health: a study protocol 

Dear Dr. Falcão:

I'm pleased to inform you that your manuscript has been deemed suitable for publication in PLOS ONE. Congratulations! Your manuscript is now with our production department. 

Kind regards, 

on behalf of

Dr. Bárbara Hatzlhoffer Lourenço 

Academic Editor

PLOS ONE